# A Diagnostic Dilemma of White Matter Lesions and Cerebral Oedema without Identifiable Cause—A Neurological Conundrum

**DOI:** 10.3390/brainsci11091238

**Published:** 2021-09-18

**Authors:** Namraj Goire, Michael Buckland, Ramesh Cuganesan, Sameer Saleem, Vivienne Lea, Roy G Beran

**Affiliations:** 1Neurology Department, Liverpool Hospital, Liverpool, NSW 2170, Australia; sameer.saleem@health.nsw.gov.au (S.S.); roy@royberan.com (R.G.B.); 2Department of Pathology, Royal Prince Alfred Hospital, Sydney, NSW 2050, Australia; michael.buckland@health.nsw.gov.au; 3Mind and Brain Institute, Sydney University, Camperdown, NSW 2050, Australia; 4Radiology Department, Liverpool Hospital, Liverpool, NSW 2170, Australia; ramesh.cuganesan@health.nsw.gov.au; 5Histopathology Department, Liverpool Hospital, Liverpool, NSW 2170, Australia; vivienne.lea@health.nsw.gov.au; 6South Western Clinical School, University of NSW, Sydney, NSW 2170, Australia; 7School of Medicine, Griffith University, Southport, QLD 4215, Australia; 8Faculty of Sociology, Sechenov Moscow First State University, 119991 Moscow, Russia

**Keywords:** white matter lesions, cerebral oedema, amnesia, neurohistophathology, diagnostic dilemma

## Abstract

Introduction: This paper describes a case of bi-frontal vasogenic oedema associated with bilateral frontal lobe and left parietal lobe white matter lesions where extensive investigations, including brain biopsy, failed to establish a diagnosis. Case Report: A 67-year-old female presented with three weeks’ history of memory loss, fatigue, insomnia, nausea, and occasional dysphasia. Physical examination was unremarkable, yet cerebral CT and MRI showed bilateral frontal lobe vasogenic oedema. Extensive investigations, including: biochemical; radiological; immunological; microbiological; haematological; histopathological; and cytological, failed to establish a confirmed diagnosis. A multidisciplinary team could not achieve a consensus for this atypical presentation. Brain biopsy was unusual, showing destructive inflammatory and subtly granulomatous disease, but an exhaustive list of auxiliary tests could not confirm a cause, and consensus favoured glial fibrillary acidic protein (GFAP) autoimmune encephalopathy. Discussion: A definitive diagnosis could not be established for this patient despite a gamut of investigations. Although some of the presenting features were consistent with GFAP astrocytopathy, initial staining of the patient’s CSF for neuronal antibodies was negative. Her symptoms and radiological changes of brain imaging improved without any corticosteroid therapy. Conclusions: Through this case report, the aim is to add to the repository of neurological sciences in the hope that future similar presentations could potentially lead to discovery of a new aetiology or contribute towards better understanding of an existing disease process.

## 1. Introduction

Cerebral oedema is often associated with trauma, intracranial space occupying lesions, vascular ischemia, or obstructive hydrocephalus [1,2,3]. Vasogenic oedema, resulting from breakdown of the blood–brain barrier, is often seen in malignant processes [1]. Several auto-immune and para-neoplastic phenomena also present with cerebral oedema and often warrant an expansive evaluation [4]. This case presentation amplifies the need for consensus diagnosis in difficult cases, complemented by exhaustive investigation, including brain biopsy. Even after an exhaustive panel of investigations, the final diagnosis may remain elusive.

## 2. Case Report

A 67-year-old Caucasian female of Lebanese ethnicity presented following a 3 week history of retrograde amnesia, intermittent emesis, lethargy, and insomnia. Her family reported occasional expressive dysphasia which spontaneously resolved. Cerebral Computed Tomography (CT), performed on presentation, showed left frontal lobe oedema with associated mild mass effect (Figure 1). Physical examination found no focal neurological deficits and she was on regular aspirin (100 mg daily) for previously diagnosed ischaemic heart disease. Her past medical history included schizophrenia, dyslipidemia, and hypertension. She migrated to Australia in 1974 and, except for a 10-year period between 1984 and 1994 when she moved back to Lebanon, she has lived in Australia. She underwent three episodes of Electro-Convulsive Therapy (ECT) between 1974 and 1994. At no time was she treated with steroids during her hospital admission.

Magnetic resonance imaging (MRI) of her brain showed bi-frontal, left parietal, and temporal vasogenic oedema with associated punctate white matter contrast enhancement (Figure 2A,B). There was nothing to suggest a space occupying lesion, and the patient was commenced on prophylactic treatment for meningo-encephalitis with BenzylPenicillin 2.4 g Q4hrly, Ceftriaxone 2 g twice daily and Aciclovir 700 mg three times a day for a total of 7 days, after undergoing a lumbar puncture. Cerebrospinal fluid (CSF) analysis showed no leucocytes, and the rest of the infectious and inflammatory screen, including meningitis panel (consisting of assays targeting nucleic acid of *Neisseria meningitidis*, *Streptococcus pneumoniae*, Varicella Zoster, Herpes simplex virus 1 and 2, and Enterovirus), mycobacteria, and fungal cultures were also negative. Although oligoclonal bands were detected in the CSF, without associated presence in serum, the patient’s clinical presentation and examination did not satisfy the diagnosis of multiple sclerosis. She also underwent central nervous system (CNS) lymphoma and autoimmune screen, including an extended panel of antibodies for limbic encephalitis that included antibodies for Neuromyelitis Optica (NMO), N-methyl-D-aspartate receptor (NMDA), and Myelin Oligodendrocyte Glycoprotein (MOG) in the CSF. Further tests were performed to provide inflammatory and infectious screening on the patient’s serum, targeting other uncommon viral, bacterial, fungal, and mycobacterial etiologies, which also returned negative results for an acute infection. Her serum was also negative for the screening of common autoimmune pathologies. 

A brain biopsy was performed and neurohistopathological analysis of the biopsy tissue showed an unusual destructive inflammatory process with a prominence of macrophages/microglia and plasma cells involving the white matter (Figure 3). The white matter was markedly oedematous and contained reactive astrocytes, as well as parenchymal and perivascular inflammation (Figure 3A). The inflammatory infiltrate consisted of numerous plasma cells (Figure 3B) (which were polyclonal on kappa and lambda staining), macrophages (Figure 3C), and lymphocytes. GFAP staining highlighted reactive astrocytes with abnormal nodularity of their processes (Figure 3D), as did aquaporin-4 staining. Of the lymphocytes, CD-3 positive T cells predominated over CD20-positive B cells (Figure 3E,F). There was no evidence of a primary demyelinating process.

In the absence of any use of steroids, during this admission, the patient had not undergone any treatment which could have raised the possibility of a partially treated CNS lymphoma. A Positron-Emission Tomograph (PET) and whole body CT scans showed no evidence of high grade lymphoma in the CNS and the only positive feature on the PET scanning was a moderately metabolically active left inguinal lymph node. This was biopsied and showed only reactive changes, presumably due to infection, with no evidence of malignancy. A progress PET scan, performed two months later, showed significantly reduced activity within this lymph node.

Further testing, to investigate for an array of rare auto-immune and typical paraneoplastic conditions, was performed. This included antibodies against CASPR2, GABA-B, DPPX, IgLON5, Amphiphysin, ANNA1 Hu, ANNA2 Ri, PCA Yo, PNMA2, MA2 Ta, CV-2 CRMP5, Recoverin, Sox-1, Titin, Zic4, GAD65, and Tr(DNER), and did not yield a positive result. Multidisciplinary teams, consisting of neurologists, radiologists, immunologists, haematologists, and histopathologists were convened to discuss this atypical presentation and, despite casting the net ‘far and wide’, the case remained a diagnostic dilemma after multiple deliberations. The consensus opinion favoured Glial Fibrillary Acid Protein (GFAP) astrocytopathy autoimmune encephalopathy, although initial staining of the CSF, for neuronal antibodies, was not supportive of the diagnosis, thereby restricting access to further testing which was not available within Australia.

The patient’s confusion, lethargy and insomnia improved over the course of her admission, and she was discharged home, with plans for outpatient follow up. An ensuing cerebral MRI, performed three months following the initial presentation, showed an interval reduction in the previously noted multifocal enhancing foci, with only subtle residual punctate foci of enhancement within the left periventricular white matter (Figure 4). The patient was recently contacted to ascertain her current status and, despite her not receiving any specific treatment to address her cerebral pathology, she currently remains well. She has not subsequently presented to a health care facility with any of the symptoms described at the outset of her complaint, and will receive ongoing follow-up in the neurology outpatients clinic.

## 3. Discussion

This case represents an unusual presentation of bilateral cerebral oedema and white matter abnormalities in a patient who initially presented with confusion, amnesia, and lethargy associated with occasional dysphasia. Vasogenic oedema and hippocampal volume increase, associated with ECT, is well described [5]. A clear correlation between this patient’s ECT, some 40 years prior to her presentation, and current imaging findings could not be established. Were there to be impugned a causal relationship between the imaging findings and the ECT, almost half a century earlier, there remains no rational explanation for the findings to be improving, as is the case in this presentation.

Extensive investigation, including tests performed on her CSF, brain biopsy, inguinal lymph node biopsy, and serum were all unremarkable, and a final diagnosis remains elusive. The consensus opinion favoured GFAP autoimmune encephalopathy, although the available tests, at the time of reporting her case, did not support that diagnosis and the definitive testing was not available in Australia.

GFAP astrocytopathy is a newer addition to the list of autoimmune conditions affecting the CNS. It was first described by Fang et al. in 2016, following a retrospective review of autoimmune profiles of patients that had presented to a neuroimmunology centre, between 1998 and 2016 [6,7]. In that study, an association was established between a novel, glial fibrillary acidic protein-specific IgG antibody, amongst a subset of patients with meningoencephalitis. This meningoencephalitis was corticosteroid-sensitive and was often associated with paraneoplastic syndromes. The IgG, detected in these patients, is the predominant intermediate filament protein in adult astrocytes, GFAPα. Since its initial description, multiple reports of GFAP astrocytopathy have been made from around the world and associated symptoms range from fever, headache, encephalopathy, and myopathy to electrolyte imbalances [8,9,10,11,12,13]. A consensus on diagnostic criteria, for GFAP astrocytopathy, awaits international agreement, and diagnosis is made following identification of GFAPα antibodies in the CSF, complemented by MRI changes [8]. A majority of GFAP patients are reported to have T2 weight changes in the MRI of the brain, although there is a dearth of reports describing vasogenic oedema associated with GFAP, as was found in this patient.

The patient’s symptoms and radiological findings have improved over the 3-month period, and this case remains a diagnostic puzzle, in the absence of definitive testing. This case raises serious implications for clinical neurology as, despite very detailed investigations, employing the latest technology available in Australia, including brain biopsy, MRI, PET, CSF analysis, and sophisticated neuro-immunological investigation, the team was unable to confirm a diagnosis. Even resorting to a multi-dimensional, multi-disciplinary consultative process, including neurologists, neuro-pathologists, neuro-radiologists, and neuro-immunologists, across three of the major teaching hospitals in Sydney and involving two Australian universities, the team could not progress the diagnosis beyond a presumptive consensus favouring GFAP encephalopathy, without definitive proof. The presentation may represent a hitherto unrecognised condition which, despite a lack of any form of aggressive intervention, is improving dramatically. It highlights the fact that, even with the major advances encountered in clinical neurology, there remains a great deal about which there is still so much to learn. It is imperative that those involved in the clinical management of difficult diagnostic problems share cases in which the final diagnosis remains a conundrum, especially where the consultative process has already been involved.

## 4. Conclusions

Through this case report, the aim is to add to the repository of neurological sciences, in the hope that future similar presentations could potentially lead to discovery of a new aetiology or contribute towards better understanding of an existing disease process.

## Figures and Tables

**Figure 1 brainsci-11-01238-f001:**
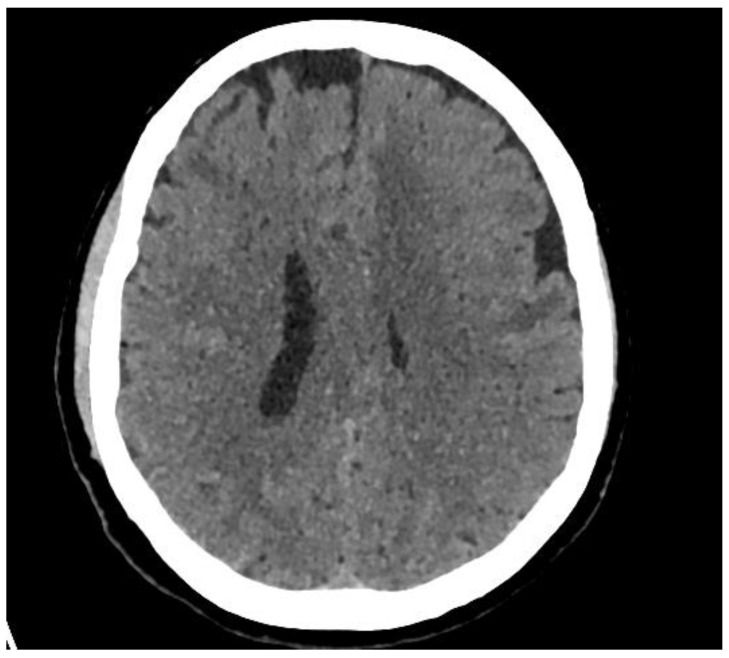
CT brain from patient’s initial presentation showing left frontal oedema with associated mild mass effect.

**Figure 2 brainsci-11-01238-f002:**
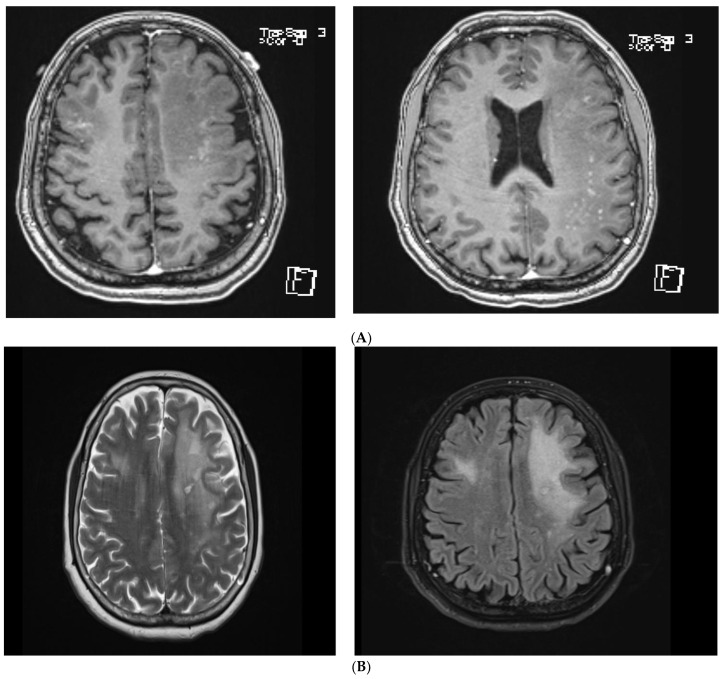
(**A**) T1 weighted images of MRI brain showing Bi-frontal, left parietal and temporal vasogenic oedema with associated punctate white matter contrast enhancement. (**B**) Corresponding T2/ FLAIR images of MRI brain showing Bi-frontal, left parietal and temporal vasogenic oedema.

**Figure 3 brainsci-11-01238-f003:**
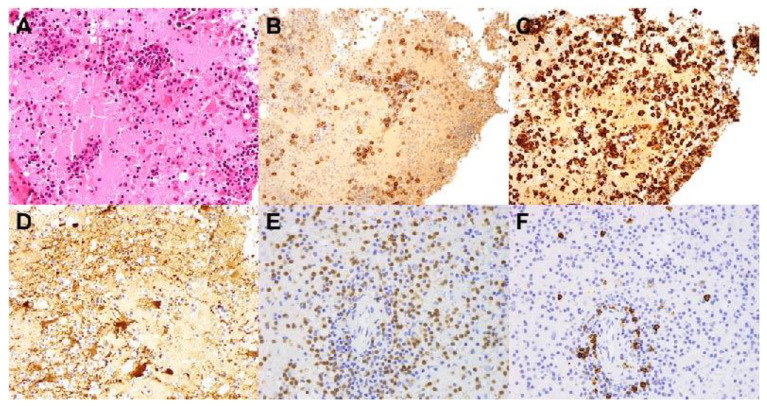
Stereotactic brain biopsy of the subcortical white matter. (**A**) Haematoxylin and eosin (H&E) stained sections show white matter oedema, perivascular and parenchymal inflammation and reactive gliosis. (**B**) CD138 immunohistochemistry highlights numerous plasma cells, and (**C**) CD163 demonstrates an abundance of macrophages/microglia. (**D**) GFAP immunohistochemistry demonstrates reactive astrocytes with unusual granularity of their processes. (**E**) Numerous CD3-positive T cells are present. (**F**) low numbers of CD20 positive B cells. (Magnification 200× A–D; 400× E,F).

**Figure 4 brainsci-11-01238-f004:**
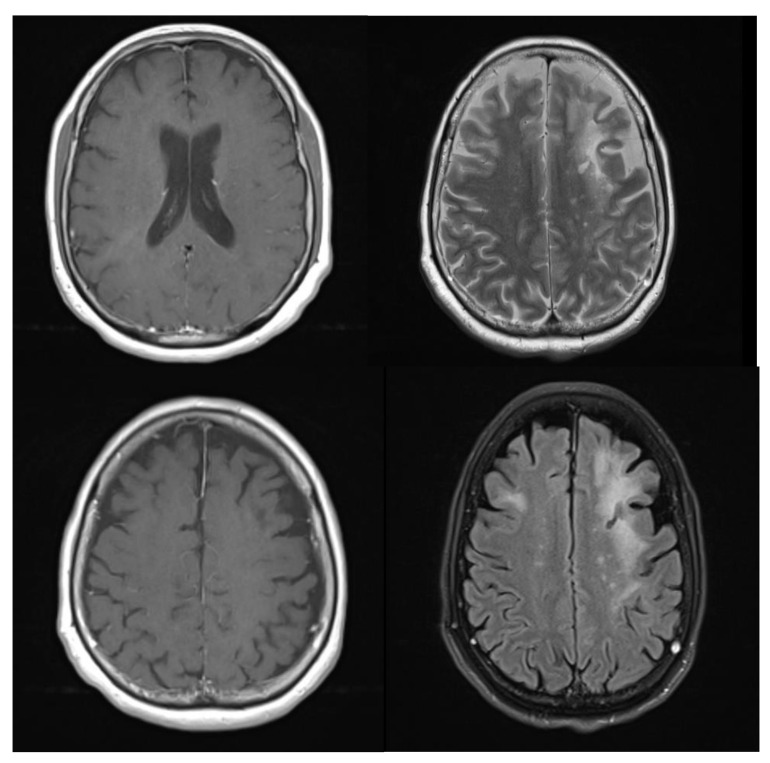
T2 FLAIR images of patient three months following intial presentation showing stable white matter vasogenic oedema/ T2 high signal. Multifocal enhancing foci are reduced in number, with only subtle residual punctate foci of enhancement within the left periventricular white matter.

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
