# Peer review of "A Diagnostic Dilemma of White Matter Lesions and Cerebral Oedema without Identifiable Cause—A Neurological Conundrum"

_brainsci, 2021, doi:10.3390/brainsci11091238_

Round 1

Reviewer 1 Report

Authors reported a case authors had difficulty for diagnosis. 

  • Though the case is well described, I would like to see not only T1-weighted MRI, but T2 or FLAIR images.
  • As of differential diagnosis, I wonder why they did not list limbic encephalitis. Authors checked NMDA receptor to exclude NMDA-receptor encephalitis, but limbic encephalitis can be caused by other factors.  It can cause edema, and prognosis is relatively well. I would like the authors to extend discussion in this respect.

Author Response

Thank you very much for your kind review. We have attempted to address your comments below.

Comment 1- Though the case is well described, I would like to see not only T1-weighted MRI, but T2 or FLAIR images.

Authors' response-Thank you, we have now added T2 and FLAIR images of patient's MRIB to the paper and they have been labelled 'image 2B'.

Comment 2- As of differential diagnosis, I wonder why they did not list limbic encephalitis. Authors checked NMDA receptor to exclude NMDA-receptor encephalitis, but limbic encephalitis can be caused by other factors.  It can cause edema, and prognosis is relatively well. I would like the authors to extend discussion in this respect.

Author's response-We agree, limbic encephalitis was certainly on the list of differential diagnoses and the patient underwent an extensive panel of tests for limbic encephalitis, all of which were not listed due to space constraints. We have mentioned our investigations for limbic encephalitis in 'Case Report' section's second paragraph, line 12 - 'She also underwent central nervous system (CNS) lymphoma and autoimmune screen, including an extended panel of antibodies for limbic encephalitis that included antibodies for Neuromyelitis Optica (NMO), N-methyl-D-aspartate receptor (NMDA), and Myelin Oligodendrocyte Glycoprotein (MOG) in the CSF.'

Reviewer 2 Report

Even though this case report is of moderate interest for the readers (I have also had one patient with unexplained cerebral edema, which remained a mystery), I unfortunately have to state that it is very poorly written. There are too many typos, some incomprehensible sentences or expressions ("a process PET scan", "reduced avidity in lymph node", "it is imperative that, those at the coal face"...), and mistakes (PET is NOT a Positron-Electron Transmission but Positron emission tomography). "An array of rare auto-immune conditions, associated with CNS lesions" is also a vague term, because the following list of antibodies is a typical paraneoplastic antibody panel, which also includes some antibodies diagnostic for autoimmune encephalitides. Case report also includes material which is quite unnecessary or redundant (such as expenses of sending CSF and serum for further analysis to USA). Case report has not been written systematically, either. All and all, I am afraid it feels like this manuscript has not been read or checked by all co-authors. 

Author Response

Thank you very much for your kind review, we have attempted to address your comments below;

Comment-Even though this case report is of moderate interest for the readers (I have also had one patient with unexplained cerebral edema, which remained a mystery), I unfortunately have to state that it is very poorly written. There are too many typos, some incomprehensible sentences or expressions ("a process PET scan", "reduced avidity in lymph node", "it is imperative that, those at the coal face"...), and mistakes (PET is NOT a Positron-Electron Transmission but Positron emission tomography). "An array of rare auto-immune conditions, associated with CNS lesions" is also a vague term, because the following list of antibodies is a typical paraneoplastic antibody panel, which also includes some antibodies diagnostic for autoimmune encephalitides. Case report also includes material which is quite unnecessary or redundant (such as expenses of sending CSF and serum for further analysis to USA). Case report has not been written systematically, either. All and all, I am afraid it feels like this manuscript has not been read or checked by all co-authors. 

Author's response- Our apologies, there does appear to have been an oversight in proof-reading the manuscript prior to submisison. The manuscript has been reviewed by Prof. Beran himself and all the typographical errors that we could identify have been removed/replaced. All of your suggestions have been taken into account/acted upon and sentences identified as vague or redundant have been replaced or removed. In particular;

  • 'Process PET scan' has been replaced with 'progress PET scan' (Case study section, paragraph 4, line 7)
  • 'Reduced avidity in the lymph node' has been replaced with 'reduced activity in the lymph node' (Case study section, paragraph 4, line 8)
  • 'It is imperative... at the coal face' has been replaced with 'It is imperative that, those involved in the clinical management of difficult diagnostic problems...' (Discussion section, paragraph 4, line 14)
  • PET is now described as Positron Emission Tomography (Case study section paragraph 4, line 3)
  • 'An array of rare auto-immune conditions, associated with CNS lesions' has now been replaced by 'Further testing, to investigate for an array of rare auto-immune and typical paraneoplastic conditions, was performed' (Case study section, paragraph 5, line 1)
  • Comment on expenses of sending CSF and serum for further analysis to USA has been removed from the text.

Round 2

Reviewer 2 Report

The contents and style of the manuscript has improved a lot and I recommend to accept the paper.